# Uncertainty Prediction for Deep Sequential Regression Using Meta Models

## Abstract

Generating high quality uncertainty estimates for sequential regression, particularly deep recurrent networks, remains a challenging and open problem. Existing approaches often make restrictive assumptions (such as stationarity) yet still perform poorly in practice, particularly in presence of real world non-stationary signals and drift. This paper describes a flexible method that can generate symmetric and asymmetric uncertainty estimates, makes no assumptions about stationarity, and outperforms competitive baselines on both drift and non drift scenarios. This work helps make sequential regression more effective and practical for use in real-world applications, and is a powerful new addition to the modeling toolbox for sequential uncertainty quantification in general.

## 1 Introduction

The ability to quantify the uncertainty of a model is one of the fundamental requirements in trusted, safe, and actionable AI (Arnold et al., 2019; Jiang et al., 2018; Begoli et al., 2019).

This paper focuses on uncertainty quantification in regression tasks, particularly in the context of deep neural networks (DNN). We define a sequential task as one involving an ordered series of input elements, represented by features, and an ordered series of outputs. In sequential *regression* tasks (SRT), the output elements are (possibly multivariate) real-valued variables. SRT occur in numerous applications, among others, in weather modeling, environmental modeling, energy optimization, and medical applications. When the cost of making an incorrect prediction is particularly high, such as in human safety, models without a reliable uncertainty estimation are perceived high risk and may not be adopted.

Uncertainty prediction in DNNs has been subject to active research, in particular, spurred by what has become known as the "Overconfidence Problem" of DNNs Guo et al. (2017), and by their susceptibility to adversarial attacks Madry et al. (2017). However, the bulk of work is concerned with non-sequential, classification tasks (see Section 2) leaving a noticeable gap for SRT.

In this paper we introduce a meta-modeling concept as an approach to achieving high-quality uncertainty quantification in DNNs for SRT. We demonstrate that it not only outperforms competitive baselines but also provides consistent results across a variety of drift scenarios. We believe the approach represents a new powerful addition to the modeling toolbox in general.

The novel contributions of this paper are summarized as follows: (1) Application of the meta-modeling concept to SRT, (2) Developing a *joint* base-meta model along with a comparison to white- and black-box alternatives, (3) Generating *asymmetric* uncertainty bounds in DNNs, and (4) Proposing a new evaluation methodology for SRT.

## 2 Related Work

Classical statistics on time series offers an abundance of work dealing with uncertainty quantification (Papoulis & Saunders, 1989). Most notably in econometrics, a variety of heteroskedastic variance models lead to highly successful application in financial market volatility analyses (Engle, 1982; Bollerslev, 1986; Mills, 1991). An Autoregressive Conditional Heteroskedastic, or ARCH, model

(Engle, 1982), and its generalized version, GARCH, (Bollerslev, 1986) are two such methods, the latter of which serves as one of our baselines.

An illuminating study (Kendall & Gal, 2017) describes an integration of two sources of uncertainty, namely the *epistemic* (due to model) and the *aleatoric* (due to data). The authors propose a variational approximation of Bayesian Neural Networks and an implicit Gaussian model to quantify both types of variability in a non-sequential classification and regression task. Based on Nix & Weigend (1994), Lakshminarayanan et al. (2017) also uses an implicit Gaussian model to improve the predictive performance of a base model, again in a non-sequential setting. Similar to Kendall & Gal (2017), the study does not focus on comparing the quality of the uncertainty to one generated by other methods. We use the implicit variance model of Kendall & Gal (2017); Oh et al. (2020); Lakshminarayanan et al. (2017), as well as the method of variational dropout of Gal & Ghahramani (2016); Kendall & Gal (2017) as baselines in our work. A meta-modeling approach was taken in Chen et al. (2019) aiming at the task of instance filtering using white-box models. The work relates to ours through the meta-modeling concept but concentrates on classification in a non-sequential setting. Besides its application in filtering, meta-modeling has been widely applied in the task of learning to learn and lifelong learning (Schmidhuber, 1987; Finn et al., 2019). However, it should be pointed out that the two applications of meta-modeling are not comparable due to their different objectives. Uncertainty in data drift conditions was assessed in a recent study (Snoek et al., 2019). The authors employ calibration-based metrics to examine various methods for uncertainty in classification tasks (image and text data), and conclude, among others that most methods' quality degrades with drift. Acknowledging drift as an important experimental aspect, our study takes it into account by testing in matched and drifted scenarios. Finally, Shen et al. (2018) described a multi-objective training of a DNN in wind power prediction, minimizing two types of cost related to coverage and bandwidth. We expand on these metrics in Section 3.3.

## 3 METHOD

### 3.1 META MODELING APPROACH

The basic concept of Meta Modeling (MM), depicted in Figure 1, involves a combination of two models comprising a *base model*, performing the main task (e.g., regression), and a *meta model*, learning to predict the base model's error behavior. Depending on the amount of information shared between these two, we distinguish several settings, namely (1) base model is a black-box (BB), (2) base is a white-box (WB, base parameters are accessible), and (3) base and meta components are trained jointly (JM). The advantages of WB and JM are obvious: rich information is available for the meta model to capture salient patterns for it to generate accurate predictions. On the other hand, the BB setting often occurs in practice and is a given.

We now formalize the MM concept as it applies to sequential regression. Let $\hat{\mathbf{y}} = F_\phi(\mathbf{x})$ be the base model function parametrized by $\phi$, where $\mathbf{x} = x_1, ..., x_N$ and $\hat{\mathbf{y}} = \hat{y}_1, ..., \hat{y}_M$ represent sequences of $N$ input feature vectors and $M$ $D$-dimensional output vectors, with $\hat{\mathbf{y}} \in \mathbb{R}^{D \times M}$. Let $\hat{\mathbf{z}} = G_\gamma(\hat{\mathbf{y}}, \mathbf{x}, \phi)$ denote the meta model, parameterized by $\gamma$, taking as input the pre-

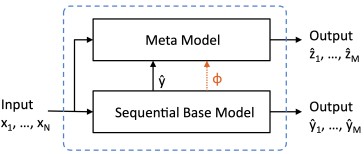

Figure 1: The concept of meta modeling

dictions, the original features, and the parameters of the base to produce a sequence of error predictions, $\hat{\mathbf{z}} \in \mathbb{R}^{D \times M}$. The parameters $\phi$ are obtained by solving an optimization problem, $\arg\min_\phi \mathbb{E}[l_b(\hat{\mathbf{y}}, \mathbf{y})]$, using a smooth loss function $l_b$, e.g., the Frobenius norm $l_b = \|\hat{\mathbf{y}} - \mathbf{y}\|_F^2$. Similarly, the parameters $\gamma$ are determined via $\arg\min_\gamma \mathbb{E}[l_m(\hat{\mathbf{z}}, \mathbf{z})] = \arg\min_\gamma \mathbb{E}[l_m(\hat{\mathbf{z}}, l_z(\hat{\mathbf{y}}, \mathbf{y}))]$ involving a loss $l_z$ quantifying the target error (residual) from the base model, and $l_m$ quantifying the prediction error of the meta model. In general, $l_b$, $l_z$, and $l_m$, may differ. The expectations are estimated using an available dataset. We used the $L_F^2$ norm for $l_b$ and $l_m$, and $L_1$ for $l_z$, as described in Section 4. Given differentiable loss functions and the DNN setting, the base and the meta model can be integrated in a single network (JM). In this case the parameters are estimated jointly via

$$\phi^*, \gamma^* = \arg\min_{\phi, \gamma} \mathbb{E}[\beta l_b(\hat{\mathbf{y}}, \mathbf{y}) + (1 - \beta) l_m(\hat{\mathbf{z}}, l_z(\hat{\mathbf{y}}, \mathbf{y}))] \tag{1}$$

whereby dedicated output nodes of the network generate $\hat{y}_t$ and $\hat{z}_t$, and $\beta$ is a hyper-parameter trading off the base with the meta loss. Thus, one part of the network tackles the base task, minimizing the base residual, while another models the residual as the eventual measure of uncertainty. As done in (Kendall & Gal, 2017), one can argue that the base objective minimizes the epistemic (parametric) uncertainty, while the meta objective captures the aleatoric uncertainty present in the data. Due to their interaction, the base loss is influenced by the estimated uncertainty encouraging it to focus on feature-space regions with lower aleatoric uncertainty. Moreover, we conjecture, the DNN base model is encouraged to encode the input in ways suitable for uncertainty quantification.

Figure 2 shows an overview of a sequential DNN architecture applied throughout our study. It includes a base encoder-decoder pair and a meta decoder connected to them. Each of these contains a recurrent memory cell - the LSTM (Hochreiter & Schmidhuber, 1997). The role of the encoder is to process the sequential input, $\mathbf{x}$, compress its information in a context vector and pass it to the base decoder. The recurrent decoder produces the regression output $\hat{\mathbf{y}}$ in $M$ time steps feeding its predictions as input in the next time steps. Evolving in time, both base LSTMs update their internal states $b_t$ and $h_t$, whereby the last state, $b_N$, serves as the context vector for the decoder. This architecture has gained wide popularity in applications such as speech-to-text (Chiu et al., 2018; Tüske et al., 2019), text-to-speech (Sotelo et al., 2017), machine translation (Sutskever et al., 2014), and image captioning (Rennie et al., 2016). Following the MM concept, we attach an additional decoder (the meta decoder) via connections to the encoder and decoder outputs. The context vector, $b_N$, is transformed by a fully connected layer (FCN in Figure 2), and both the $\hat{y}_t$ output as well as the internal state, $h_t$, are fed into the meta component. As mentioned above, the meta decoder generates uncertainty estimates, $\hat{z}_t$.

Given the architecture depicted in Figure 2, we summarize the three settings as follows:(1) Joint Model (JM): parameters are trained according to Eq. (1) with certain values of $\beta$. (2) White-Box model (WB): base parameters $\phi$ are trained first, followed by parameters $\gamma$, also accessing $\phi$. (3) Black-Box model (BB): same as (2) without access to $\phi$.

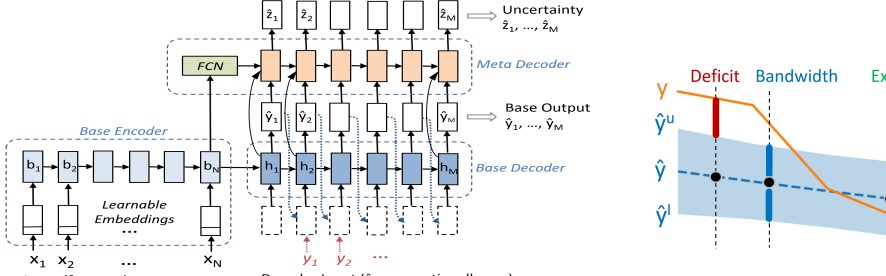

Figure 3: Bandwidth, excess, and deficit costs.

Figure 2: Encoder-Decoder architecture integrating a base and a meta model

**Generating Symmetric and Asymmetric Bounds**  The choice of loss function, $l_z$, gives rise to two scenarios. If $l_z$ is an even function, e.g., $l_z(\hat{\mathbf{y}}, \mathbf{y}) = \|\hat{\mathbf{y}} - \mathbf{y}\|_1$, the meta-model targets $\mathbf{z}$ capture base error equally in both directions: above and below the target. Hence, the uncertainty $\hat{\mathbf{z}}$ predicted at test time will represent a interval symmetric around $\hat{y}$. If, on the other hand, $l_z$ takes the sign in $z$ into account, it is possible to dedicate separate network nodes $\gamma_l, \gamma_u \in \gamma$ to capturing lower and upper band estimates, $\hat{\mathbf{z}}_l$ and $\hat{\mathbf{z}}_u$, respectively, thus accomplishing asymmetric prediction. Let $\boldsymbol{\delta} = \hat{\mathbf{y}} - \mathbf{y}$. For the asymmetric scenario the meta objective is modified as follows:

$$\gamma^* = \arg\min_{\gamma} \mathbb{E}[l_m(\mathbf{z}_l, \max\{\boldsymbol{\delta}, 0\}) + l_m(\mathbf{z}_u, \max\{-\boldsymbol{\delta}, 0\})] \tag{2}$$

## 3.2 Baselines

**Implicit Heteroskedastic Variance**  Lakshminarayanan et al. (2017); Kendall & Gal (2017); Oh et al. (2020) applied a Gaussian model $\mathcal{N}(\mu, \sigma^2)$ to the output of a neural network predictor, where $\mu$

represents the prediction and $\sigma^2$ its uncertainty due to observational (aleatoric) noise. The model is trained to minimize the negative log-likelihood (NLL), with the variance being an implicit uncertainty parameter (in that it is trained indirectly) which is allowed to vary across the feature space (heteroskedasticity). We apply the Gaussian in the sequential setting by planting it onto the base decoder's output (replacing the meta decoder) and train the network using the NLL objective: $\phi^* = \arg\min_\phi \mathbb{E}\left[\sum_{t=1}^{M}\sum_{d=1}^{D} \frac{(\hat{y}_{t,d}-y_{t,d})^2}{\sigma_{t,d}^2} + \log\sigma_{t,d}^2\right]$ with $D$ output nodes modeling the regression variable, $\hat{y}_t$, and separate D output nodes modeling the $\log\sigma_t^2$, at time $t$.

**Variational Dropout**   Gal & Ghahramani (2016) established a connection between dropout (Srivastava et al., 2014), i.e., the process of randomly omitting network connections, and an approximate Bayesian inference. We apply the variational dropout method to the base encoder and decoder. By performing multiple runs per test sequence, each with a different random dropout pattern, the base predictions are calculated as the mean and the base uncertainty as the variance over such runs. This Bayesian method, along with the variational approximation, captures the parametric (epistemic) uncertainty of the model, hence it fundamentally differs from the Gaussian model as well as our proposed approach.

**GARCH Variance**   Introduced in (Bollerslev, 1986; Engle, 1982), the Generalized Autoregressive Conditional Heteroskedastic (GARCH) variance model belongs among the most popular statistical methods. A GARCH(p,q) assumes the series to follow an autoregressive moving average model and estimates the variance at time $t$ as a linear combination of past $q$ residual terms, $\epsilon^2$, and $p$ previous variances, $\sigma^2$: $\sigma_t^2 = \alpha_0 + \sum_{i=1}^{q}\alpha_i\epsilon_{t-i}^2 + \sum_{i=0}^{p}\beta_i\sigma_{t-i}^2$.

The $\alpha_0$ term represents a constant component of the variance. The parameters, $\alpha, \beta$ are estimated via maximum-likelihood on a training set. The GARCH process relates to the concept in Figure 1 in that it acts as the meta-model predicting the squared residual. We use the GARCH as a baseline only on one of the datasets for reasons discussed in Section 4.

**Constant-Band Baseline**   A consistent comparison of uncertainty methods is difficult due to the fact that each generates an uncertainty around different base predictions. Therefore, as a reference we also generate a constant symmetric band around each base predictor. Such a bound represents a homoskedastic process – a sensible choice in many well-behaved sequential regression problems, corresponding to a GARCH(0,0) model. We will use this reference point to compute a relative gain of each method as explained in Section 3.3.

## 3.3   Evaluation Methodology

**Core Metrics**   Unlike with classification tasks, where standard calibration-based metrics apply (Snoek et al., 2019), we need to consider two aspects arising in regression, roughly speaking: (1) what is the extent of observations falling outside the uncertainty bounds (Type 1 cost), and (2) how excessive are the bounds (Type 2 cost). An optimal bound captures all of the observation while being least excessive in terms of its bandwidth. Shen et al. (2018), among others, defined two measures reflecting these aspects (miss rate and bandwidth) which we adopt below (Eqs. (3) and (4)) while adding two refinements (Eqs. (5) and (6)). Let $\hat{\mathbf{y}}^l = \hat{\mathbf{y}} - \hat{\mathbf{z}}_l$ and $\hat{\mathbf{y}}^u = \hat{\mathbf{y}} + \hat{\mathbf{z}}_u$ denote the predicted lower and upper bound, respectively. Recall that $\hat{\mathbf{y}} \in \mathbb{R}^{D\times M}$. We define the following metrics:

$$\text{Missrate}(\hat{\mathbf{y}}^l, \hat{\mathbf{y}}^u, \mathbf{y}) = 1 - \frac{1}{MD}\sum_{d,t:y_{dt}\in[\hat{y}_{dt}^l,\hat{y}_{dt}^u]} 1 \tag{3}$$

$$\text{Bandwidth}(\hat{\mathbf{y}}^l, \hat{\mathbf{y}}^u, \mathbf{y}) = \frac{1}{2MD}\sum_{d=1}^{D}\sum_{t=1}^{M} \hat{y}_{dt}^u - \hat{y}_{dt}^l \tag{4}$$

$$\text{Excess}(\hat{\mathbf{y}}^l, \hat{\mathbf{y}}^u, \mathbf{y}) = \frac{1}{MD}\sum_{d,t:y_{dt}\in[\hat{y}_{dt}^l,\hat{y}_{dt}^u]} \min\left\{y_{dt} - \hat{y}_{dt}^l, \hat{y}_{dt}^u - y_{dt}\right\} \tag{5}$$

$$\text{Deficit}(\hat{\mathbf{y}}^l, \hat{\mathbf{y}}^u, \mathbf{y}) = \frac{1}{MD}\sum_{d,t:y_{dt}\notin[\hat{y}_{dt}^l,\hat{y}_{dt}^u]} \min\left\{|y_{dt} - \hat{y}_{dt}^l|, |y_{dt} - \hat{y}_{dt}^u|\right\} \tag{6}$$

Table 1: Overview statistics of the MITV and the SPE9PR datasets

| Dataset | Total Samples | Input Features | Output Dim | Time Resol. | Partition Size | | | |
|---------|---------------|----------------|------------|-------------|---------------|---------|----------|--------|
| | | | | | TRAIN | DEV | DEV2 | TEST |
| MITV | 48204 | 8 | 1 | 1 hr | 33744 | 4820 | 4820 | 4820 |
| SPE9PR | 28000 $\times 100$ | 248 | 4 | 90 days | 24000 $\times 100$ | 1000 $\times 100$ | 1000 $\times 100$ | 1000 $\times 100$ |

Figure 3 illustrates these metrics. The relative proportion of observations lying outside the bounds (miss rate) ignores the *extent* of the bound's short fall. The Deficit, Eq. (6), captures this. The type 2 cost is captured by the Bandwidth, Eq. (4). However, its range is indirectly compounded by the underlying variation in $\hat{\mathbf{y}}$ and $\mathbf{y}$. Therefore we propose the Excess measure, Eq. (5), which also reflects the Type 2 cost, but just the portion above the minimum bandwidth necessary to include the observation.

**Calibration**   In general, DNNs offer few guarantees about the behavior of their output. DNNs tend to produce miscalibrated classification probabilities (Guo et al., 2017). In order to evaluate the uncertainty across models, it is necessary to establish a common operating point (OP). We achieve this via a scaling calibration. For symmetric bounds, we assume that $\mathbf{y} = \hat{\mathbf{y}} + \hat{\mathbf{z}} \odot \boldsymbol{\epsilon}$ where $\boldsymbol{\epsilon} \in \mathbb{R}^{D \times M}$ is a random i.i.d. matrix and $\hat{\mathbf{z}}$ is the predicted non-negative uncertainty band. Let $Z_{dt} = \frac{y_{dt} - \hat{y}_{dt}}{\hat{z}_{dt}}$. Using a held-out dataset, we can obtain an empirical distribution in each output dimension: $\text{cdf}(\{Z_{dt}\}_{1 \leq t \leq M}), d = 1, ..., D$. It is then possible to find the value $\varepsilon_d^{(p)}$ for a desired quantile $p$, e.g., $\varepsilon_d^{(0.95)}$, and construct the prediction bound at test time: $\left[\hat{y}_{dt} - \hat{z}_{dt}\varepsilon_d^{(p)}, \hat{y}_{dt} + \hat{z}_{dt}\varepsilon_d^{(p)}\right]$. Assuming $Z_d$ is stationary, in expectation, this bound will contain the desired proportion $p$ of the observations, i.e., $\mathbb{E}[\text{Missrate}(\hat{\mathbf{y}}^l, \hat{\mathbf{y}}^u, \mathbf{y})] = 1 - p$.

This scaling is applied in our evaluation to compare the excess, bandwidth and deficit at fixed miss rates as well as setting a minimum cost OP (see Section 3.3). An Algorithm to find a scale factor for a desired value of any of the four metrics in $O(M^2)$ operations is given in the Appendix.

**Metrics Used in Reporting**   The following OP-based measures are used in reporting: (1) Excess, Deficit, Bandwidth at a fixed Missrate, averaged over Missrate $= \{0.1, 0.05, 0.01\}$, and (2) Minimum Excess-Deficit cost, where cost $= \frac{1}{2}(\text{Excess} + \text{Deficit})$ with the minimum found over all calibrations (OPs).

For each system and measure, $m_s$, a symmetric constant-band baseline, $m_{fixed}$, is also generated and a relative gain with respect to this reference calculated: $gain_s = 100 \times \frac{m_{fixed} - m_s}{m_{fixed}}\%$. Finally, the error rate of the base predictor is calculated as $E_{base}(\hat{\mathbf{y}}, \mathbf{y}) = \frac{1}{D} \sum_{d=1}^{D} \frac{\|\hat{y}_d - y_d\|_1}{\|y_d\|_1}$, where $\hat{y}_d, y_d$ are $d$-th row vectors of $\hat{\mathbf{y}}, \mathbf{y}$.

# 4   EXPERIMENTS

**Datasets**   Two sequential regression datasets, namely the *Metro Interstate Traffic Volume (MITV)* dataset[1] and the *SPE9 Reservoir Production Rates (SPE9PR)* dataset[2], were experimented with. Both originate from real-world applications, involve sequential input/output variables, and provide for scenarios with varying degrees of difficulty.

The MITV dataset is a collection of hourly weather features with the target of the regression being the hourly traffic volume, recorded by the Minnesota DoT continuously between 2012 and 2018. The SPE9PR, on the other hand, is a large collection of mathematical simulations of a reservoir field with varying input and output sequences with each simulation comprising a sequence with 100 time steps. The regression targets in this case are multivariate and correspond to field production rates.

---

[1] https://archive.ics.uci.edu/ml/machine-learning-databases/00492/
[2] https://developer.ibm.com/technologies/artificial-intelligence/data/oil-reservoir-simulations

The SPE9PR also contains a data partition collected under distributional drift. Full detail on both datasets including their preprocessing can be found in the Appendix.

**Training Procedure** Each dataset was partitioned into TRAIN, DEV, DEV2, and TEST sets (with SPE9PR also providing a TEST-drift set), as listed Table 1. While the TRAIN/DEV partitions served basic training, the DEV2 was used in determining hyperparameters and operating points (calibration). The TEST sets were used to produce the reported metrics. While a single sample in the SPE9PR represents a complete sequence of 100 steps, the MITV data come as a single contiguous sequence. The partitioning of the MITV set is strictly ordered by time, whereby the DEV sequence follows TRAIN, DEV2 follows DEV, and TEST follows DEV2. After partitioning, each MITV sequence was processed by a sliding window of length 36 hours (in 1-hour steps). This resulted in a series of $(n - 35) \times 36$ subsequences ($n$ denotes the partition size) to feed the encoder-decoder model. When testing on the MITV, DNN predictions from such sliding windows were recombined into a contiguous prediction sequence again.

The base encoder-decoder network (see Figure 2) is trained using the Adam optimizer (Kingma & Ba, 2014) with a varying initial learning rate, `lr`, in two stages: (1) Training of all parameters using TRAIN while providing the ground truth as the decoder input at each time step. (2) Building on the previous, the training continues, however, decoder predictions from step $t - 1$ are fed as decoder inputs at step $t$—a mode referred to as *emulation* by Bengio et al. (2015). All hyperparameter values are listed in the Appendix.

JOINT MODEL, SYMMETRIC (JMS), AND ASYMMETRIC (JMA): The common training steps are performed using the objective in Eq. (1), with $\beta = 1.0$, first. Then, the joint training continues with $\beta = 0.5$ as long as the objective improves on DEV. In a final step, the model switches to using DEV as training with $\beta = 0.0$ until no improvement on TRAIN is seen. A similar procedure is followed for the JMA, except using Eq. (2).

WHITE-BOX MODEL, SYMMETRIC (WBMS): The basic training is performed with $\beta = 1.0$. Next, only meta model parameters are estimated using the DEV/TRAIN sets with $\beta = 0.0$.

BLACK-BOX MODEL, SYMMETRIC (BBMS): Base training is performed. The base model processes the DEV set to generate residual $\mathbf{z}$. A separate encoder-decoder model is then trained using $(\mathbf{x}, \mathbf{z})$.

JOINT MODEL WITH VARIANCE (JMV): The two common steps are performed using the NLL objective (see Section 3.2) with the variance-related parameters first fixed, and, in a subsequent step, allowing the variance parameters to be adjusted, until convergence. This is to aid stability in training (Nix & Weigend, 1994).

DROPOUT MODEL SYMMETRIC (DOMS): Dropout with a rate of 0.25 (inputs, outputs) and 0.1 (internal states), determined as best performing on the DEV2 set, were applied in both the base encoder and the decoder. The two common steps were performed. At test time, the model was run 10 times per each test sequence to obtain the mean prediction and standard deviation.

GARCH: The GARCH model from Section 3.2 is used with $p = q = 5$. The lag value was determined using an autocorrelation chart showing attenuation at lags $> 5$. We only apply this baseline to the MITV dataset as it provides a contiguous time series. The model parameters were trained using the DEV2 partition.

Throughout the experiments, the size of each LSTM cell was kept fixed at 32 for the base encoder/decoder, and at 16 for the meta decoder. The base sizing has been largely driven by our preliminary study showing it suffices in providing accurate base predictions.

**Testing Procedure** As mentioned above the SPE9PR dataset has two TEST partitions: one for a matched and one for a drifted condition. While the MITV dataset does not provide an explicit source of drift, we induce drift by creating a discrepancy in the modeling procedure between training and test: In the non-drift condition, the DNN's decoder is given access to the past 12 hours worth of traffic observations to make a forecast for the next 24 hours. This is achieved by spanning a 36-hour window and feeding the decoder inputs the first 12 hours of ground truth, during training. Now to create the drift scenario we test the MITV model without providing those first 12 hours of observations and the model uses their own predictions for that period instead. This emulates a "model drift" condition in that the model, trained to rely on actual observations, is getting its own noisy predictions.

Table 2: Relative optimum and cross-validated gains ($G^*$, $G^x$) using the Excess-Deficit metrics. $E_{base}$ denotes the base predictor's error. Within each column, elements marked[†] are in a statistical tie, all other values are mutually significant at $p < 0.01$

| System | MITV | | | | | | SPE9PR | | | | | |
| | **match** | | | **Model Drift** | | | **Match** | | | **Data Drift** | | |
| | $E_{base}$ | $\%G^*$ | $\%G^x$ | $E_{base}$ | $\%G^*$ | $\%G^x$ | $E_{base}$ | $\%G^*$ | $\%G^x$ | $E_{base}$ | $\%G^*$ | $\%G^x$ |
|---|---|---|---|---|---|---|---|---|---|---|---|---|
| JMS | .155[†] | **57.8**[†] | **59.3**[†] | .187[†] | **52.5** | **55.6** | .165 | **46.5** | **46.6** | .291 | **56.6** | **50.5** |
| WBMS | .159[†] | **53.8**[†] | **55.3**[†] | .190[†] | 39.4 | 42.3 | .162 | 44.8 | 44.9 | .313 | 54.5 | 43.8 |
| JMV | .167 | 20.4 | 25.5 | .179 | 20.0[†] | 17.5[†] | .159 | 45.3 | 45.5 | .334 | -6.5 | 4.4 |
| DOMS | .144 | 13.2 | 12.8 | .155 | 16.6[†] | 17.3[†] | .177 | 1.3 | 1.4 | .279 | -0.2 | -5.2 |
| BBMS | .153[†] | -0.4 | 1.2 | .188[†] | -8.9 | -3.0 | .170 | 31.9 | 30.3 | .326 | 11.6 | 12.8 |
| GARCH | .155[†] | -1.8 | 3.1 | .187[†] | -14.7 | -4.0 | n/a | n/a | n/a | n/a | n/a | n/a |

Table 3: Relative optimum and cross-validated gains ($G^*$ and $G^x$) on Bandwidth and Excess-Deficit metrics for the asymmetric JMA model.

| | Evaluation | MITV | | SPE9PR | |
| | | **Match** | **Drift** | **Match** | **Drift** |
|---|---|---|---|---|---|
| | JMA Base Error | .158 | .200 | .168 | .320 |
| Bandwidth | JMA, $\%G^*$ | **44.1** | **37.0** | **35.5** | **54.3** |
| | JMA, $\%G^x$ | 33.1 | 26.4 | 35.5 | 8.5 |
| | JMS, $\%G^*$ | 37.5 | 31.6 | 29.1 | 34.3 |
| | JMS, $\%G^x$ | 35.2 | 27.2 | 28.7 | -4.4 |
| Ex-Deficit | JMA, $\%G^*$ | 45.3 | 29.0 | 30.9 | 50.8 |
| | JMA, $\%G^x$ | 40.4 | 28.6 | 30.8 | 24.2 |
| | JMS, $\%G^*$ | 57.8 | 52.5 | **46.5** | **56.6** |
| | JMS, $\%G^x$ | **59.3** | **55.6** | 46.6 | 50.5 |

**Results with Symmetric Bounds**    Table 2 compares the proposed symmetric-bounds systems (JMS, WBMS, BBMS) with the baselines (JMV, DOMS, GARCH). The relative error of the base predictor is given in the $E_{base}$ column. The uncertainty quality is reported in Table 2 is the average gain in excess-deficit metrics, as defined in Section 3.3. Columns labeled as $G^*$ contain measurements made at an operating point (OP) determined on the test set itself, while those labeled as $G^x$ use an OP from a held-out (DEV2) set. While $G^x$ reflects generalization of the calibration, $G^*$ values are interesting as they reveal the potential of each method. Based on a paired permutation test (Dwass, 1957) all but entries marked with † are mutually significant at $p < 0.01$.

From Table 2 we make the following observations: (1) the JMS model dominates all other models across all conditions. The fact that it outperforms the WBMS indicates there is a benefit to the joint training setup, as conjectured earlier. (2) The WBMS dramatically outperforms the BBMS model, which remains only as good as a constant band for MITV data, indicating it is hard to reliably predict residuals from only the input features. (3) The most competitive baseline is the JMV model. As discussed in Section 3.2, the JMV shares some similarity with the meta-modeling approach. (4) The JMS and WBMS models perform particularly well in the strong drift scenario (SPE9PR), suggesting that white-box features play an essential role in achieving generalization. Finally, (5) the DOMS model works well on MITV data but provides no benefit in the SPE9PR, which could be due the aleatoric uncertainty playing a dominant role in this dataset. In almost all cases, however, the averaging of base predictions in DOMS results in lowest error rates of the base predictor.

Representative samples of JMS and JMV uncertainty bounds are shown in Figure 4 (MITV) and Figure 5 (SPE9PR). They illustrate a clear trend in the results, namely that the JMS (also seen with WBMS) model are better able to cover the actual observation, particularly when the base prediction tends to make large errors. Additional plots can be found in the Appendix, and also a notebook to visualize all test samples is provided as part of the Supplementary Material.

**Results with Asymmetric Bounds**    Generating asymmetric bounds is a new intriguing aspect of DNN-based meta-models. Using the JMA model, we first recorded the accuracy with which the

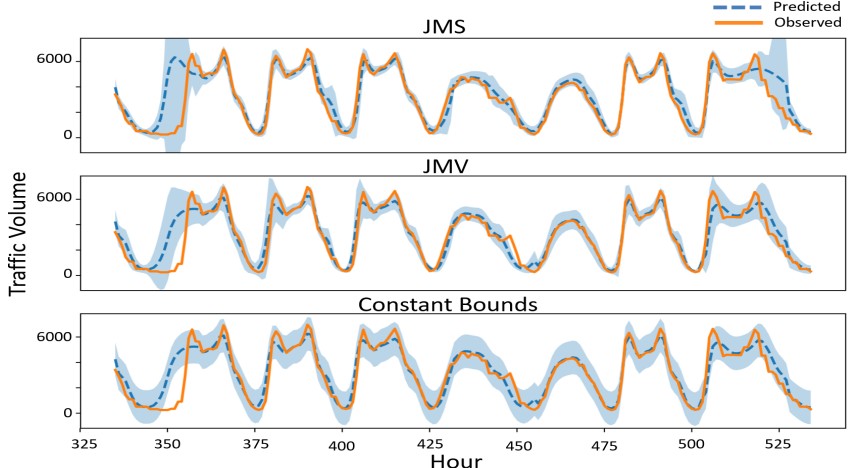

Figure 4: Sample of traffic volume predictions with uncertainty generated by the JMS and JMV models, along with a constant bound (around JMV) (miss rate set to 0.1 on TEST).

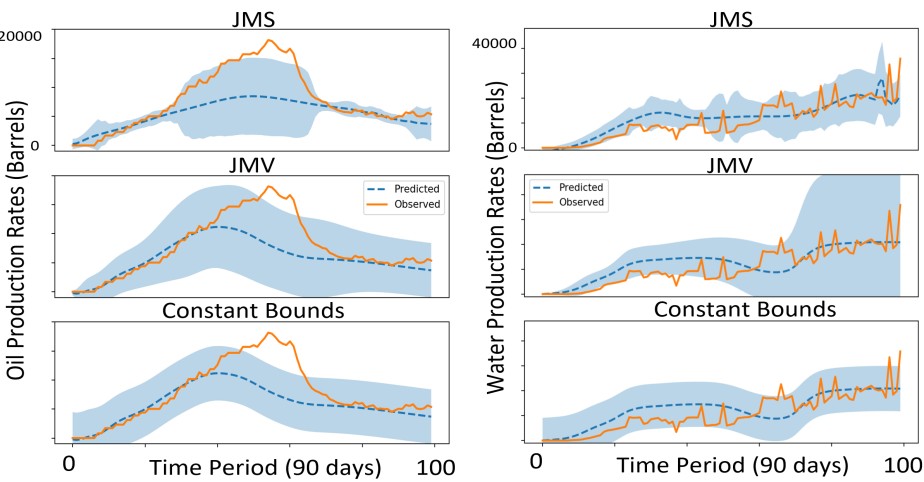

Figure 5: SPE9PR samples of oil (left) and water (right) production rates ("drift" scenario, miss rate set to 0.1 on TEST).

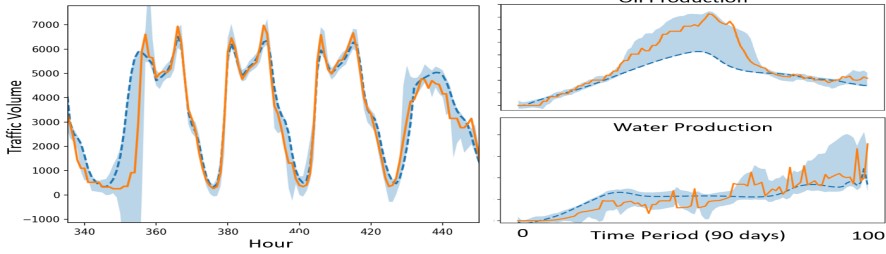

Figure 6: Samples of asymmetric bounds produced by the JMA. MITV sample (left) and SPE9PR (right) correspond to segments shown in Figure 4 and 5.

asymmetric output agrees *in sign* (orientation) with the observed base discrepancy. Averaged over each of the two datasets, this accuracy is at 83.3%, and 91.1%. The promise of asymmetric bounds lies in its potential to reduce the *bandwidth* cost. Since the Excess and Deficit metrics ignore the absolute bandwidth, we also evaluate the JMA model using the Bandwidth metric (Eq. (4)), averaged over the same OPs. The results are shown in Table 3 comparing the JMA model to the best symmetric model, JMS. The JMS model outperforms JMA in all scenarios on Excess-Deficit, however, compared on the bandwidth metric, the JMA dominates benefiting from its orientation capability. Upon visual inspection the output of the JMA is appreciably better in bandwidth: Figure 6 shows samples on both datasets. In most instances the bounds behave as expected, expending the bulk of bandwidth in the correct direction. An interesting question arises whether it is possible to utilize the asymmetric output as a correction on the base predictor. Our preliminary investigation shows that a naive combination leads to degradation in the base error, however, this question remains of interest for future work.

## 5 CONCLUSIONS

In this work we demonstrated that meta-modeling (MM) provides a powerful new framework for uncertainty prediction. Through a systematic evaluation of the proposed MM variants we report considerable relative gains over a constant reference baseline and show that they not only outperform all competitive baselines but also show stability across drift scenarios. A jointly trained model integrating the base with a meta component fares best, followed by a white-box setup, indicating that trainable white-box features play an essential role in the task. Besides symmetric uncertainty, we also investigated generating asymmetric bounds using dedicated network nodes and showed their benefit in reducing the uncertainty bandwidth. We believe these results open an exciting new research avenue for uncertainty quantification in sequential regression.

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

---

**Algorithm 1** Find best scale for a given metric value

---

**Input:** Observation, base and meta predictions $\{\hat{y}_t, y_t, \hat{z}_t^l, \hat{z}_t^u\}_{1 \leq t \leq M}$; metric function $f$; target value $\rho^*$
**Output:** Scale factor $\varepsilon^*$
**for** $t \leftarrow 1$ **to** $M$ **do**
    $\delta_t \leftarrow \hat{y}_t - y_t$.
    $\varepsilon_t \leftarrow \begin{cases} \frac{\delta_t}{\hat{z}_t^l} & \text{for } \delta_t \geq 0 \\ \frac{-\delta_t}{\hat{z}_t^u} & \text{otherwise} \end{cases}$
    **for** $k \leftarrow 1$ **to** $M$ **do**
        $\hat{y}_k^l \leftarrow \hat{y}_k - \varepsilon_t \hat{z}_k^l$
        $\hat{y}_k^u \leftarrow \hat{y}_k - \varepsilon_t \hat{z}_k^u$
    **end for**
    $\rho_t \leftarrow f\left(\left\{\hat{y}_k^l, \hat{y}_k^u, y_k\right\}_{1 \leq k \leq M}\right)$
**end for**
$t^* \leftarrow \arg\min_t |\rho_t - \rho^*|$
$\varepsilon^* \leftarrow \varepsilon_{t^*}$

---

## A    ALGORITHM TO FIND A SCALING FACTOR

Section 3.3 discusses the scaling calibration in the context of the four metrics: Missrate, Bandwidth, Excess, and Deficit. Algorithm 1 finds a scale factor for a desired value of any of these four metrics in $O(M^2)$ operations.

## B    ADDITIONAL DATASET AND IMPLEMENTATION DETAILS

### B.1    DATASETS

#### B.1.1    METRO INTERSTATE TRAFFIC VOLUME (MITV)

The dataset is a collection of hourly westbound-traffic volume measurements on Interstate 94 reported by the Minnesota DoT ATR station 301 between the years 2012 and 2018. These measurements are aligned with hourly weather features[3] as well as holiday information, also part of the dataset. The target of regression is the hourly traffic volume. This dataset was released in May, 2019.

The MITV input features were preprocessed to convert all categorical features to trainable vector embeddings, as outlined in Figure 2. All real-valued features as well as the regression output were standardized before modeling (with the test predictions restored to their original range before calculating final metrics). Overall dataset statistics are listed in Table 1 and further processing steps are given in Section 4.

#### B.1.2    MITV PREPROCESSING

As described in Section B.1.1 and Table 1, the MITV dataset comes with 8 input features, among which 3 are categorical. Here we list the relevant parsing and encoding steps used in our setup. The raw time stamp information was parsed to extract additional features such as day of the week, day of the month, year-day fraction, etc. Table 4 shows the corresponding list. Standardization was performed on the input as well as output, as per Table 4, whereby the model predictions were transformed to their original range before calculating final metrics.

#### B.1.3    SPE9 RESERVOIR PRODUCTION RATES (SPE9PR)

This dataset originates from an application of oil reservoir modeling. A reservoir model (RM) is a space-discretized approximation of a geological region subject to modeling. Given a sequence of drilling actions (input), a physics-based PDE-solver (simulator) is applied to the RM to generate

---

[3]provided by OpenWeatherMap

Table 4: MITV Input and Output Specifications

| Feature Name | Range | Categorical | Embedding Dimension | Standardized | Final Dimension |
|---|---|---|---|---|---|
| INPUT | | | | | |
| `day_of_month` | integer $\in [0, 30]$ | Y | 3 | N | 3 |
| `day_of_week` | integer $\in [0, 6]$ | Y | 3 | N | 3 |
| `month` | integer $\in [0, 11]$ | Y | 3 | N | 3 |
| `frac_yday` | real $\in [\frac{1}{365}, 1]$ | N | - | Y | 1 |
| `weather_type` | integer $\in [0, 10]$ | Y | 3 | N | 3 |
| `holiday_type` | integer $\in [0, 11]$ | Y | 3 | N | 3 |
| `temperature` | real $\in \mathbb{R}$ | N | - | Y | 1 |
| `rain_1h` | real $\in \mathbb{R}_0^+$ | N | - | Y | 1 |
| `snow_1h` | real $\in \mathbb{R}_0^+$ | N | - | Y | 1 |
| `clouds_all` | real $\in [0, 100]$ | N | - | Y | 1 |
| Total | | | | | 20 |
| OUTPUT | | | | | |
| `traffic_volume` | real $\in \mathbb{R}_0^+$ | N | - | Y | 1 |
| Total | | | | | 1 |

sequences of future production rates (oil, gas, water production), typically over long horizons Killough (1995). The objective is to train a DNN and accurately predict outputs on unseen input sequences. We used the publicly available SPE9[4] RM, considered a reference for benchmarking reservoir simulation in the industry, and an open-source simulator[5] to produce 28,000 simulations, each with 100 randomized actions (varying type, location, and control parameters of a well) inducing production rate sequences over a span of 25 years, in 90-day increments, i.e., 100 time steps. Furthermore, the RM was partitioned into two regions, A and B. While most of the actions are located in the region A, we also generated 1000 sequences with actions located in the region B thus creating a large degree of mismatch between training and test. The test condition in region B will be referred to as "drift" scenario.

### B.1.4 SPE9PR PreProcessing

The Table 5 lists details on the SPE9PR features (also refer to Section B.1.3 and Table 1). The SPE9PR dataset contains input sequences of actions and output sequences of production rates. An action (feature `type_of_well`), at a particular time, represents a decision whether to drill, and if so, what type of well to drill (an injector or a producer well), or not to drill (encoded by "0"), hence the cardinality is 3. In case of a drill decision, further specifications apply, namely the x- and y-location on the surface of the reservoir, local geological features at the site, and well control parameters. There are 15 vertical cells in the SPE9 each coming with 3 geological features (rel. permeability, rel. porosity, rock type), thus the local geology is a 45-dimensional feature vector at a particular $(x, y)$ location. Finally, every well drilled so far may be controlled by a parameter called "Bottom-Hole Pressure" (BHP). Since we provision up to 100 wells of each of the two types, a 200-dimensional vector arises containing BHP values for these wells at any given time. Standardization was performed on the input as well as output as specified in Table 5 whereby the model predictions were transformed to their original range before calculating and reporting final metrics.

---

[4]https://github.com/OPM/opm-data/blob/master/spe9/SPE9.DATA
[5]https://opm-project.org/

Table 5: SPE9PR Input and Output Specifications

| Feature Name | Range | Categorical | Embedding Dimension | Standardized | Final Dimension |
|---|---|---|---|---|---|
| INPUT | | | | | |
| type_of_well | integer $\in \{0, 1, 2\}$ | Y | 3 | N | 3 |
| location_x | integer $\in [0, 24]$ | Y | 10 | N | 10 |
| location_y | integer $\in [0, 25]$ | Y | 10 | N | 10 |
| vertical_geology | real $\in \mathbb{R}^{45}$ | N | - | Y | 45 |
| per_well_control | real $\in \mathbb{R}^{200}$ | N | - | Y | 200 |
| Total | | | | | 258 |
| OUTPUT | | | | | |
| oil_prod_field_rate | real $\in \mathbb{R}_0^+$ | N | - | Y | 1 |
| gas_prod_field_rate | real $\in \mathbb{R}_0^+$ | N | - | Y | 1 |
| water_prod_field_rate | real $\in \mathbb{R}_0^+$ | N | - | Y | 1 |
| water_inj_field_rate | real $\in \mathbb{R}_0^+$ | N | - | Y | 1 |
| Total | | | | | 4 |

Table 6: Hyperparameter settings

| Hyper-parameter | Where used | Value | Comment |
|---|---|---|---|
| Learning rate | all | 0.001 | Stage 1 training |
| Learning rate | all | 0.0002 | Stage 2 training, see Section 4 |
| Batch size | all | 100 | |
| $L_2$ penalty coefficient | all, except DOMS | 0.0001 | |
| $L_2$ penalty coefficient | DOMS | 0.0 | |
| Dropout | DOMS | 0.25/0.1/0.25 | LSTM Input/State/Output (encoder and decoder) |
| Base LSTM size | all | 32 | |
| Meta LSTM size | meta models | 16 | |
| $\beta$ in Eq. (1) | 1.0/0.5/0.0 | see Section 4 | |

## B.2 TRAINING SETUP

### B.2.1 HYPERPARAMETERS

Hyperparameters have been determined in two ways: (1) learning rate, regularization, batch, and LSTM size were adopted from an unrelated experimental study performed on a modified reservoir SPE9 (Anonymized), (2) We used DEV2 to determine the dropout rates in the DOMS model. The value $\beta = 0.5$ was chosen ad-hoc (as a midpoint between pure base and pure meta loss) without further optimization.

## B.3 IMPLEMENTATION NOTES

All DNNs were implemented in Tensorflow 1.11. Training was done on a Tesla K80 GPU, with total training time ranging between 3 (MITV) and 24 (SPE9PR) hours. The GARCH Python implementation provided in the arch library was used.

Table 7: Relative optimum and cross-validated gains ($G^*$, $G^{xval}$) for the MITV dataset, using the Excess-Deficit metrics. $E_{base}$ denotes the base predictor's error. Within each column, elements marked[†] are in a statistical tie, all other values are mutually significant at $p < 0.01$

| System | MITV | | | | | |
|--------|------|------|------|------|------|------|
| | **match** | | | **drift** | | |
| | $E_{base}$ | $\%G^*$ | $\%G^{xval}$ | $E_{base}$ | $\%G^*$ | $\%G^{xval}$ |
| JMS | .155[†] | **57.8**[†] | **59.3**[†] | .187[†] | **52.5** | **55.6** |
| WBMS | .159[†] | **53.8**[†] | **55.3**[†] | .190[†] | 39.4 | 42.3 |
| JMV | .167 | 20.4 | 25.5 | .179 | 20.0[†] | 17.5[†] |
| DOMS | .144 | 13.2 | 12.8 | .155 | 16.6[†] | 17.3[†] |
| BBMS | .153[†] | -0.4 | 1.2 | .188[†] | -8.9 | -3.0 |
| GARCH | .155[†] | -1.8 | 3.1 | .187[†] | -14.7 | -4.0 |

Table 8: Relative optimum and cross-validated gains ($G^*$, $G^{xval}$) for the SPE9PR dataset, using the Excess-Deficit metrics. $E_{base}$ denotes the base predictor's error. Within each column, elements marked[†] are in a statistical tie, all other values are mutually significant at $p < 0.01$

| System | SPE9PR | | | | | |
|--------|--------|------|------|------|------|------|
| | **match** | | | **drift** | | |
| | $E_{base}$ | $\%G^*$ | $\%G^{xval}$ | $E_{base}$ | $\%G^*$ | $\%G^{xval}$ |
| JMS | .165 | **46.5** | **46.6** | .291 | **56.6** | **50.5** |
| WBMS | .162 | 44.8 | 44.9 | .313 | 54.5 | 43.8 |
| JMV | .159 | 45.3 | 45.5 | .334 | -6.5 | 4.4 |
| DOMS | .177 | 1.3 | 1.4 | .279 | -0.2 | -5.2 |
| BBMS | .170 | 31.9 | 30.3 | .326 | 11.6 | 12.8 |

## C  ADDITIONAL RESULTS

### C.1  INDIVIDUAL METRICS

For a more detailed view of the averages in Table 2, we show a split by the individual metrics in Table 9, and, for the SPE9PR which has a total of four output variables, a split by the individual variables in Table 10.

### C.2  ADDITIONAL VISUALIZATIONS

In addition to the sample visualizations shown in Section 2 for the JMS, JMV, and JMA systems, here we show same sections of the data and visualize output of all systems. Figures 7 and 8 show the first simulation in the test set of the SPE9PR dataset for the drift and non-drift condition and all its output components, respectively. Figures 9 and 10 show the output on the MITV drift and non-drift condition, respectively. For each model, a miss rate value of 0.1 across the entire test set was used in the visualizations.

#### C.2.1  INTERACTIVE NOTEBOOK

We also provide an interactive notebook that allows for inspecting all system output on an arbitrary portion of the test data in both the non-drift and drift condition. Please refer to the `README` file within the `zip`-file uploaded as the Supplementary Material part of our submission.

#### C.2.2  JMV VARIANCE TUNING ON DEV DATA

Section 4 lists individual training steps for each system. It is noted that the meta-modeling arrangements have used the DEV partition for tuning in a final step. The motivation for using a partition not included in training the base model is the avoidance of meta-training on biased targets, i.e., targets generated by the base model on its own training data. In this context, a question arises whether a sim-

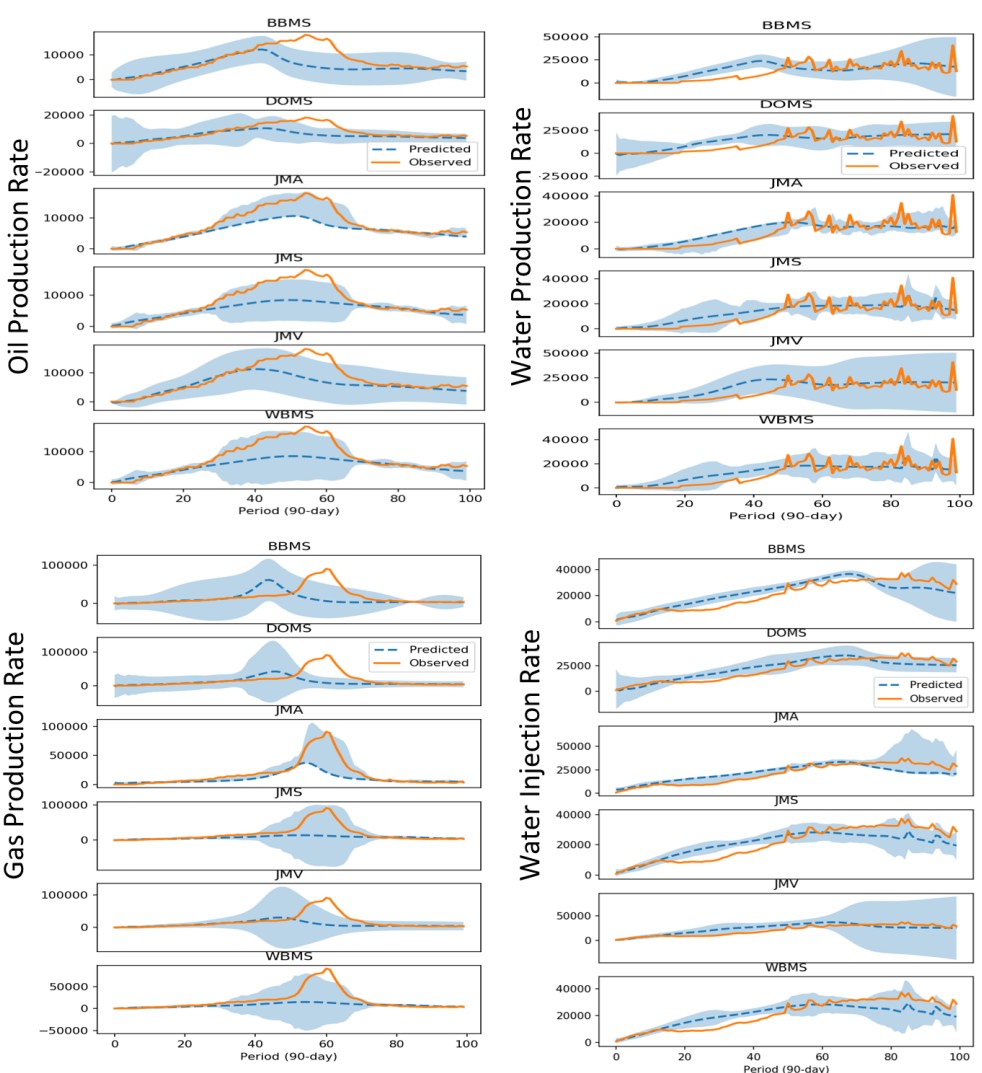

Figure 7: Sample from SPE9PR (simulation 0, drift condition), all components and systems shown.

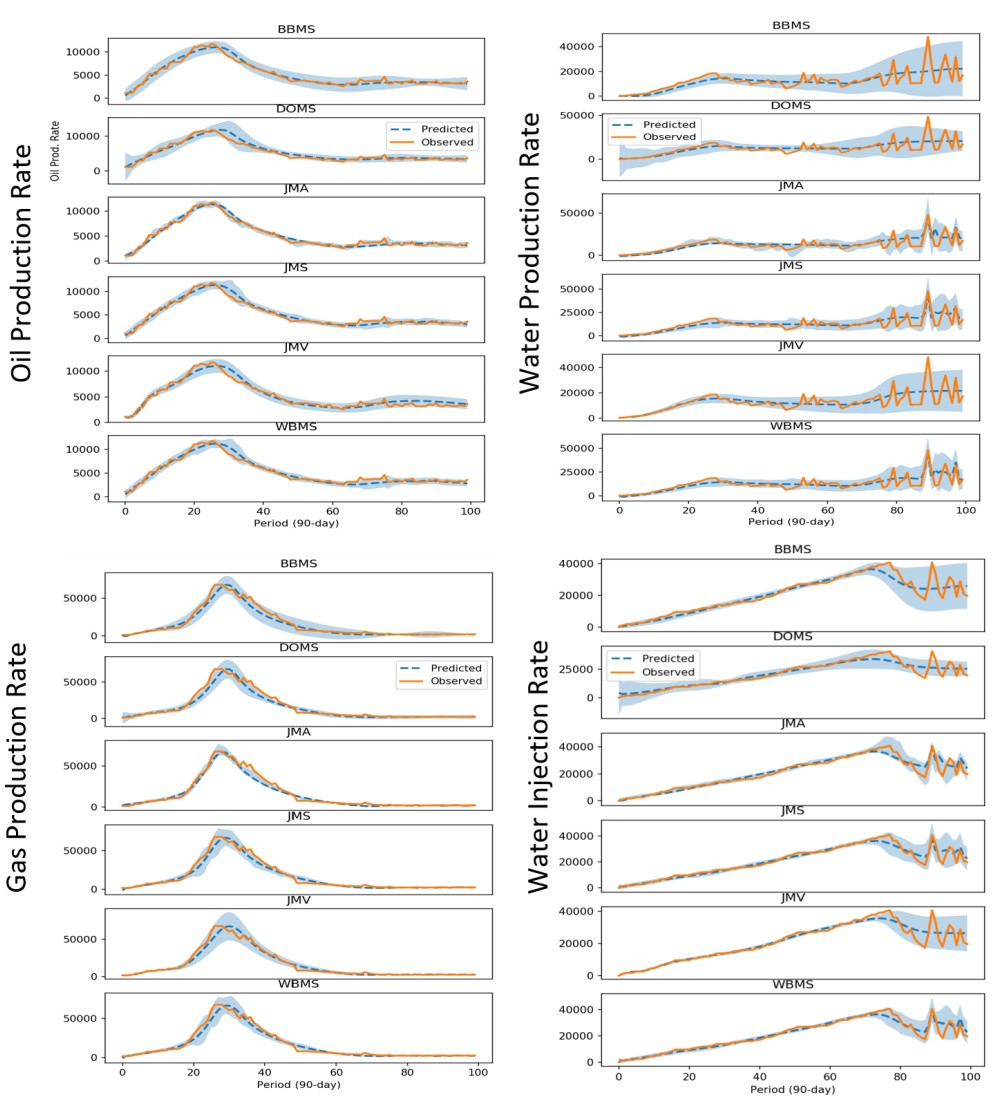

Figure 8: Sample from SPE9PR (simulation 0, non-drift condition), all components and all systems shown.

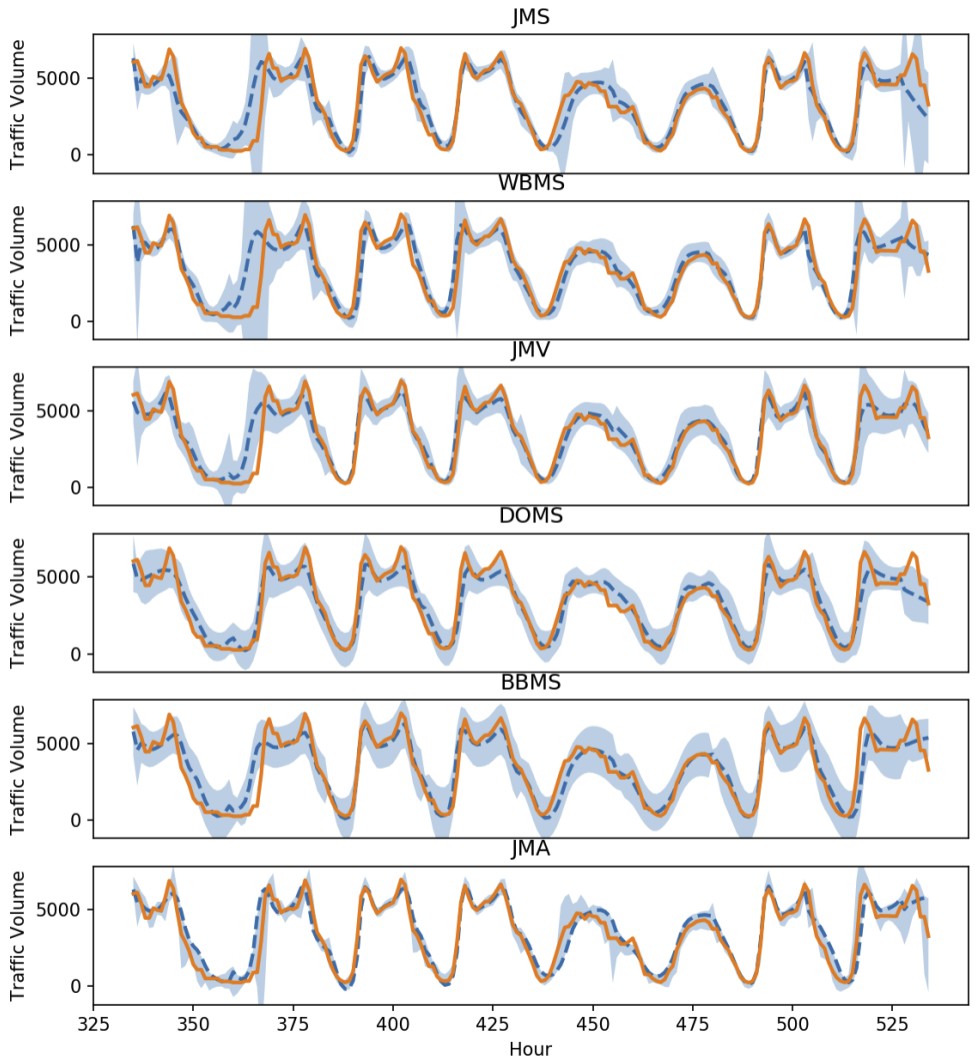

Figure 9: Sample from MITV (drift condition), all systems shown.

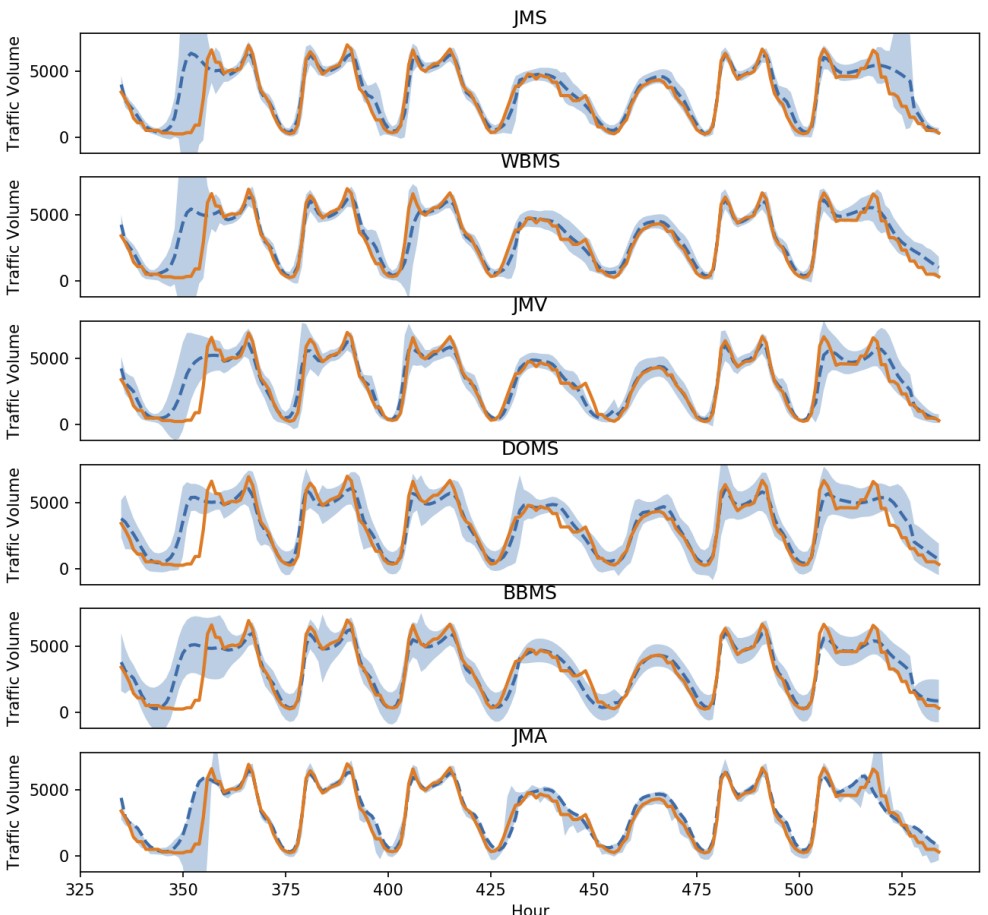

Figure 10: Sample from MITV (non-drift condition), all systems shown.

ilar tuning step could help the JMV model. We followed the training steps described in Section 4 and then updated the network nodes tied to the variance parameter while keeping the rest of the network fixed. The Table 11 shows the results on the MITV dataset. It seems the benefit of the tuning step does not materialize. In all but the cross-validated drift case the gain decreases (albeit insignificantly) when applying the DEV-only tuning. We conjecture that the benefit of the tuning step exists with the meta-model because of the direct supervision of the meta-model's prediction. In contrast, the variance in the JMV setting is learned implicitly and may not suffer from the "biased-target" problem mentioned above.

Table 9: Symmetric gains split by individual metric (compare to Table 2)

| Model | Deficit@0.01 | Deficit@0.05 | Deficit@0.1 | Excess@0.01 | Excess@0.05 | Excess@0.1 | MinCost | Average |
|---|---|---|---|---|---|---|---|---|
| $\%G^*$, MITV "Non-Drift" Scenario | | | | | | | | |
| JMS | 61.3 | 74.8 | 75.6 | 64.5 | 54.6 | 40.1 | 33.9 | 57.8 |
| WBMS | 48.0 | 73.8 | 74.3 | 62.7 | 50.7 | 35.1 | 32.3 | 53.8 |
| JMV | -33.9 | 31.9 | 32.5 | 35.8 | 31.4 | 28.2 | 16.8 | 20.4 |
| DOMS | 28.1 | 24.8 | 23.1 | 11.5 | 3.3 | -1.6 | 2.8 | 13.1 |
| BBMS | 9.7 | 27.9 | 16.3 | -18.8 | -18.7 | -18.7 | -0.8 | -0.4 |
| $\%G^{xval}$, MITV "Non-Drift" Scenario | | | | | | | | |
| JMS | 75.9 | 78.1 | 77.4 | 58.0 | 51.9 | 39.6 | 33.9 | 59.3 |
| WBMS | 67.4 | 79.6 | 77.7 | 57.7 | 42.7 | 29.8 | 32.3 | 55.3 |
| JMV | 4.4 | 38.9 | 35.8 | 29.1 | 27.1 | 26.3 | 16.8 | 25.5 |
| DOMS | 78.2 | 63.9 | 62.3 | -35.7 | -32.4 | -49.2 | 2.8 | 12.8 |
| BBMS | 61.6 | 21.5 | 15.4 | -61.0 | -12.5 | -15.9 | -0.9 | 1.2 |
| $\%G^*$, MITV "Drift" Scenario | | | | | | | | |
| JMS | 40.5 | 78.0 | 77.5 | 57.7 | 45.0 | 31.9 | 36.7 | 52.5 |
| WBMS | 18.4 | 68.8 | 70.8 | 43.6 | 32.8 | 15.6 | 25.8 | 39.4 |
| JMV | -3.8 | 29.0 | 30.9 | 23.0 | 25.5 | 21.6 | 13.8 | 20.0 |
| DOMS | 23.4 | 23.7 | 24.7 | 11.6 | 14.2 | 11.4 | 7.3 | 16.6 |
| BBMS | -38.2 | 5.7 | 8.7 | -11.8 | -12.2 | -14.1 | -0.2 | -8.9 |
| $\%G^{xval}$, MITV "Drift" Scenario | | | | | | | | |
| JMS | 71.7 | 81.6 | 80.1 | 53.2 | 39.1 | 27.1 | 36.7 | 55.6 |
| WBMS | 67.9 | 79.2 | 78.5 | 22.9 | 19.4 | 2.0 | 25.8 | 42.3 |
| JMV | -27.2 | 28.5 | 32.7 | 29.1 | 25.9 | 19.6 | 13.8 | 17.5 |
| DOMS | 38.4 | 18.9 | 23.4 | 2.8 | 16.6 | 13.3 | 7.4 | 17.3 |
| BBMS | 6.2 | 13.2 | 13.1 | -17.5 | -17.4 | -18.6 | -0.3 | -3.0 |
| $\%G^*$, SPE9PR "Non-Drift" Scenario | | | | | | | | |
| JMS | 55.7 | 55.8 | 56.8 | 46.1 | 42.5 | 37.7 | 30.8 | 46.5 |
| WBMS | 50.9 | 54.5 | 55.6 | 47.4 | 41.5 | 35.4 | 28.5 | 44.8 |
| JMV | 50.5 | 54.2 | 53.7 | 51.8 | 43.3 | 37.0 | 26.4 | 45.3 |
| DOMS | 7.1 | 10.2 | 10.4 | -6.9 | -5.8 | -6.8 | 1.0 | 1.3 |
| BBMS | 51.0 | 52.6 | 50.0 | 25.1 | 18.2 | 10.6 | 15.7 | 31.9 |
| $\%G^{xval}$, SPE9PR "Non-Drift" Scenario | | | | | | | | |
| JMS | 56.8 | 56.5 | 57.0 | 45.7 | 42.0 | 37.4 | 30.8 | 46.6 |
| WBMS | 50.7 | 54.9 | 56.2 | 47.5 | 41.2 | 34.9 | 28.5 | 44.8 |
| JMV | 51.4 | 56.0 | 55.2 | 51.6 | 42.3 | 35.5 | 26.4 | 45.5 |
| DOMS | 7.7 | 9.5 | 10.1 | -7.0 | -5.3 | -6.4 | 1.0 | 1.4 |
| BBMS | 76.4 | 61.2 | 54.8 | -11.5 | 10.6 | 5.0 | 15.7 | 30.3 |
| $\%G^*$, SPE9PR "Drift" Scenario | | | | | | | | |
| JMS | 65.4 | 69.0 | 68.4 | 52.6 | 50.0 | 46.3 | 44.5 | 56.6 |
| WBMS | 63.7 | 68.0 | 66.9 | 53.5 | 48.1 | 41.8 | 39.5 | 54.5 |
| JMV | 29.2 | 22.7 | 21.9 | -33.9 | -42.8 | -52.9 | 10.3 | -6.5 |
| DOMS | -1.4 | 9.4 | 12.0 | -10.7 | -4.5 | -8.5 | 2.7 | -0.1 |
| BBMS | 19.4 | 33.8 | 31.0 | -7.6 | 1.1 | -3.0 | 6.7 | 11.6 |
| $\%G^{xval}$, SPE9PR "Drift" Scenario | | | | | | | | |
| JMS | 95.0 | 85.5 | 78.7 | 22.8 | 18.8 | 14.4 | 38.5 | 50.5 |
| WBMS | 94.7 | 84.1 | 77.3 | 16.8 | 5.0 | -5.9 | 34.9 | 43.8 |
| JMV | -40.0 | -25.2 | -12.2 | 48.3 | 31.6 | 16.9 | 11.3 | 4.4 |
| DOMS | 16.4 | 10.1 | 9.1 | -23.4 | -23.5 | -26.9 | 1.8 | -5.2 |
| BBMS | 40.5 | 9.4 | 7.9 | 0.8 | 17.3 | 9.0 | 5.0 | 12.8 |

Table 10: Symmetric gains split by individual components - SPE9PR only (compare to Table 2). OPR=Oil Production Rate, WPR=Water Production Rate, GPR=Gas Production Rate, WIN=Water Injection Rate.

| Model | $E_{base}$ OPR | WPR | GPR | WIN | Average | Excess-Deficit OPR | WPR | GPR | WIN | Average |
|---|---|---|---|---|---|---|---|---|---|---|
| %$G^*$, SPE9PR "Non-Drift" Scenario | | | | | | | | | | |
| JMS | 0.12 | 0.28 | 0.17 | 0.09 | 0.17 | 18.25 | 66.45 | 53.60 | 47.61 | 46.48 |
| WBMS | 0.12 | 0.28 | 0.17 | 0.09 | 0.16 | 17.78 | 62.08 | 52.43 | 47.03 | 44.83 |
| JMV | 0.10 | 0.29 | 0.16 | 0.09 | 0.16 | 6.67 | 74.50 | 43.74 | 56.24 | 45.29 |
| DOMS | 0.12 | 0.31 | 0.18 | 0.11 | 0.18 | -6.88 | 0.68 | 21.64 | -10.16 | 1.32 |
| BBMS | 0.12 | 0.30 | 0.16 | 0.10 | 0.17 | -3.85 | 74.44 | 16.89 | 40.08 | 31.89 |
| %$G^*$, SPE9PR "Non-Drift" Scenario | | | | | | | | | | |
| JMS | 0.12 | 0.28 | 0.17 | 0.09 | 0.17 | 19.13 | 66.98 | 52.18 | 48.19 | 46.62 |
| WBMS | 0.12 | 0.28 | 0.17 | 0.09 | 0.16 | 18.32 | 62.60 | 51.07 | 47.39 | 44.85 |
| JMV | 0.10 | 0.29 | 0.16 | 0.09 | 0.16 | 6.92 | 75.74 | 43.12 | 56.23 | 45.50 |
| DOMS | 0.12 | 0.31 | 0.18 | 0.11 | 0.18 | -6.56 | 0.46 | 21.94 | -10.27 | 1.39 |
| BBMS | 0.12 | 0.30 | 0.16 | 0.10 | 0.17 | -3.84 | 76.80 | 7.71 | 40.63 | 30.33 |
| %$G^*$, SPE9PR "Drift" Scenario | | | | | | | | | | |
| JMS | 0.28 | 0.30 | 0.45 | 0.13 | 0.29 | 49.97 | 24.13 | 119.73 | 32.54 | 56.59 |
| WBMS | 0.31 | 0.34 | 0.45 | 0.14 | 0.31 | 51.49 | 27.09 | 102.64 | 36.77 | 54.50 |
| JMV | 0.30 | 0.32 | 0.51 | 0.20 | 0.33 | 1.12 | -6.18 | 26.99 | -47.96 | -6.51 |
| DOMS | 0.28 | 0.28 | 0.45 | 0.11 | 0.28 | -20.24 | -6.97 | 22.32 | 4.31 | -0.15 |
| BBMS | 0.32 | 0.31 | 0.54 | 0.13 | 0.33 | -4.18 | -1.04 | 26.64 | 25.13 | 11.64 |
| %$G^{xval}$, SPE9PR "Drift" Scenario | | | | | | | | | | |
| JMS | 0.28 | 0.30 | 0.45 | 0.13 | 0.29 | 61.81 | 39.10 | 74.23 | 27.01 | 50.54 |
| WBMS | 0.31 | 0.34 | 0.45 | 0.14 | 0.31 | 59.11 | 34.21 | 60.51 | 21.51 | 43.84 |
| JMV | 0.30 | 0.32 | 0.51 | 0.20 | 0.33 | 2.15 | 9.15 | 19.51 | -13.28 | 4.38 |
| DOMS | 0.28 | 0.28 | 0.45 | 0.11 | 0.28 | 0.81 | -27.32 | 15.73 | -10.09 | -5.22 |
| BBMS | 0.32 | 0.31 | 0.54 | 0.13 | 0.33 | 2.55 | 4.67 | 20.74 | 23.36 | 12.83 |

Table 11: Relative optimum and cross-validated gains using Excess-deficit metrics for the JMV system without (JMV) and with (JMV-$\sigma$) variance tuning on DEV. Elements marked[†] within same column are in a statistical tie.

| Evaluation | %$G^*$ | %$G^x$ |
|---|---|---|
| JMV (Drift) | 20.0 | 17.5[†] |
| JMV-$\sigma$ (Drift) | 15.0 | 19.4[†] |
| JMV (Match) | 20.4[†] | 25.5 |
| JMV-$\sigma$ (Match) | 16.2[†] | 16.9. |

