# OpenReview forum: "Uncertainty Prediction for Deep Sequential Regression Using Meta Models"
_ICLR.cc/2021/Conference — Reject_

### Official Review · AnonReviewer3 · 2020-10-28
**Meta-modeling is a strong method to estimate uncertainty in sequential tasks**

**Rating:** 7
**Confidence:** 3

**Review:**

In this work, the authors present a meta-modeling approach to provide predictions with uncertainty estimates in a sequential task. They develop a white box, black box, and joint modeling method that allows them to apply their method to a variety of scenarios. These methods differ based on the amount of information provided to a meta-learner which has the goal of predicting errors $\hat{z} \in \mathbb{R}^{D \times M} $. The authors also incorporate the ability to make asymmetric uncertainty bounds. They apply this method and many baeslines to two datasets: MITV and SPE9PR.

Strengths:
* Strong presentation and emphasis of the different components of the white box, black box, and joint methods
* Good spectrum of baselines
* Comparison on drifting data

Weaknesses:
* Many of the figures seem compressed by the $\LaTeX $ -- consider remaking your figures with the appropriate aspect ratio
* The tables seem like they could refined a little more (i.e. Table 3) -- I think I may have missed the definition of "xval" somewhere in the work and "match" and "drift" could be capitalized.
* Could you address how this work could be applied to other methods such as RNN's?

Ultimately, I think this is a strong work that establishes an exciting method for uncertainty estimation. Because of it's strong presentation, novelty, and experiments, I rate this as a clear accept.

---

### Official Review · AnonReviewer1 · 2020-10-28
**The paper needs more work**

**Rating:** 5
**Confidence:** 3

**Review:**

This paper describes a method to generate symmetric and asymmetric uncertainty estimates. The method is proposed to work for the non-stationarity processes found in real-world applications. The paper introduces a meta-modelling concept as an approach to achieve high-quality uncertainty quantification in deep neural networks for sequential regression tasks. The paper also introduces metrics for evaluating the proposed approach. A proposed meta-modelling approach is related to the work of Chen et al. (2019) which is mainly used for classification task in a non-sequential setting, however, the proposed method is mainly for the sequential setting.

The paper has interesting explorations for handling uncertainty and its evaluation. However, the paper needs more work for clarity and rigorous analysis. For example, equation 2 is not clear to me and needs explanation and/or related citations. For instance, why even function reflects symmetric uncertainty and how separating network nodes produces lower and upper band estimates to accomplish asymmetric prediction.

The evaluation metrics were introduced (Eq: 3,4,5,6) to measure the effectiveness of the proposed method. Is there any existing evaluation metric for symmetric and asymmetric uncertainty estimates that can be used for measuring the effectiveness of the proposed method?

There are two proposals in this paper: one is a meta-modelling concept for uncertainty measurement and another is the evaluation metrics to evaluate the uncertainty. It will be better to separate them, the first part needs to be evaluated with exiting uncertainty evaluation metrics with some well-known benchmark datasets. This part is to measure the effectiveness of the meta-modelling concept using the existing metric. The second part can be for proposing new evaluation metric and justify the reason behind the proposed the new metrics.

Two real datasets was used for experimentation which is a good idea for evaluating on real applications. For the purpose of evaluating the proposed systems, it is also necessary to utilize some benchmark datasets from the literature.

Minor comment: In figure 1, change the subscript N to M for consistency with the text.

---

### Official Review · AnonReviewer4 · 2020-10-29
**A simple approach to uncertainty estimation, but I have some questions about experimental validation**

**Rating:** 6
**Confidence:** 4

**Review:**

In this paper, the authors propose a technique for uncertainty estimation in regression with neural networks. The basic idea is to use an auxiliary "meta model" that (in the authors' best performing setting) has access to the base model and is trained jointly with it. The purpose of the meta model is to predict the error characteristics of the base model, which of course naturally leads to error bars. In order to account for the fact that the models errors on the train set are unlikely to be representative, the authors make use of a validation set on which the meta model is trained further.

The general idea of using a meta model to predict the error characteristics of a base model potentially has some advantages over existing approaches, including a fairly easy approach to modelling asymmetric error bars and only requiring one forward pass through the full model at test time. I do have some concerns about the experimental evaluation.

A critical gap here is a lack of comparison to a relatively large literature on *explicit* ensembling and (recently) efficient explicit ensembling techniques (e.g., BatchEnsemble) for uncertainty calibration that have worked very well across a variety of complex domains including computer vision, natural language processing and sequence modelling, and model based reinforcement learning. For purely neural network based approaches, many would argue that these ensembling techniques represent the current state of the art.

Some details of the experimental setup were a little unclear. Since the datasets hard a time component, how were DEV and DEV2 sets chosen relative to test? Can the authors confirm that the TRAIN set always occurred strictly before the DEV and TEST sets in time, and that the DEV sets occurred strictly before the TEST set in time? To be frank, some of the uncertainty intervals in 1D are shockingly good if this is pure extrapolation, with the uncertainty estimates seeming to always vary almost perfectly with the true error. If this were a consistent feature of the model, particularly with asymmetric uncertainty outputs, I would imagine it would be straightforward to dramatically improve mean predictive performance as well using information from the meta model, which seems surprising.

How much of the benefit is due to the additional tuning on a dev set? For example, suppose you tuned a standard heteroskedastic NN's variance outputs on a dev set? Why the decision to focus specifically on sequence modelling? I think the inclusion of WBMS and BBMS as training settings is largely unnecessary, to be frank, in particular the inclusion of BBMS. I think it would be very rare that we would need to augment a blackbox model with uncertainty estimates rather than having the ability to train from scratch to produce a well calibrated model.

I am a little surprised to see the invention of new calibration metrics for regression but not the inclusion of existing metrics for this that have been used in statistics for a long time. Metrics like continuous ranked probability scores do not assume a particular distributional form of output and are well studied and likely to be highly correlated with some of these more slightly ad hoc metrics. The bandwidth "metric" is also a little strange as a metric, as it doesn't depend on the true label y and rather just encourages confidence generally.

On a minor note, most of the plots in the paper suffer from being scaled manually after being generated, which has lead to all of the text being unnaturally stretched.

--------
After author feedback, I feel that the authors did address a number of my points, including one or two that were indeed addressed in the text that I must have simply missed. Therefore, I am improving my score.

The one point if any that I continue to disagree with the authors on is that variational dropout is sufficient to cover the space of explicit ensembling methods. Many of these "cost effective" ensembling approaches still continue to compare to direct explicit ensembling with the very explicit goal of having a baseline that is known to work well but removed from the computational cost concerns. I feel that this comparison should be standard practice.

---

### Official Review · AnonReviewer2 · 2020-10-29
**Review for "Uncertainty Prediction for Deep Sequential Regression Using Meta Models"**

**Rating:** 5
**Confidence:** 4

**Review:**

**Overview**

This paper applies "meta-modeling" to the time sequence regression task.
The idea is to meta-learn a base model that not only fits the labels, but is also easier for a meta-model to learn uncertainty quantification. In practice, the paper proposes to jointly train two models: one predicts labels, and another predicts the residuals (can be interpreted as uncertainty measurement). The two loss terms are $\beta$ weighted and summed together, as in traditional multi-task learning setup. Several variances are evaluated on two datasets: MITV and SPE9PR.

**Pros**

The authors proposed and studied 3 variances: 1) jointly train base and meta model's weights $\phi,\gamma$ with the regression target and the residual term; 2, 3) sequentially train base and meta model's $\phi,\gamma$ with/without parameter access of the base model's $\phi$.
The evaluation results on MITV and SPE9PR suggests that Jointly training approach out-performs other variances and is more robust in the presence of dataset and model drift.  The evaluation metrics consists of Missrate, Bandwidth, Excess and Deficit (Fig 3), which are well designed for the sequence regression and uncertainty prediction task.

**Cons**

The main idea of the paper is to utilize meta-modeling to jointly train a meta model to estimate the error behavior of the base model.  However "meta-modeling" is not a new concept, and it is usually referred to as meta-learning and learning-to-learn in the literature.  The later normally requires compute the gradient of gradient (or with its first-order approximation, like MAML and Reptile) or directly predicting weights of a neural network. The proposed approach doesn't really fit into this category, as  the weight of base and meta model $\phi,\gamma$ are separately learnt in a multi-task fashion (e.g., there are 2 tasks in this setup: one is the regression task for the base model and the other is the residual prediction task for the meta model).

I also think predicting the asymmetric residual bound loss term (eq 2) might confuse the meta-model. Would meta-model's predicted upper bound still be meaningful if it thinks the prediction is less than the regression target?

The two evaluation datasets are on limited domains. The second one is from a physics-based simulator instead of from real-world data collection. Have the author consider doing evaluation on a more broader domains to demonstrate the proposed approach can bring consistent gains?

Lastly, since the drift are of different nature for the two eval dataset (data drift vs model drift), perhaps worth using different names in Table 2 (data drift for SPE9PR and model drift for MITV)


**Reason for the decision**

This paper attempts to introduce meta-learning to the sequence regression and uncertainty prediction task. The evaluation results are encouraging among a few variances and baselines, and the metrics are well thought of.  However the main reason for me to give marginally reject is that: 1. the approach doesn't really fit into the meta-learning category and 2. the two evaluation datasets are on somewhat limited domains.

---

### Decision · Program_Chairs · 2021-01-07
**Final Decision**

**Decision:**

Reject

**Comment:**

This is a borderline paper. Initially, it received  weaker reviews (than the current ones) and after rebuttal, one reviewer slightly increased the review rating.  Among 4 reviews, there is one clearly positive one.

Among negative reviews, there are repeated concerns about evaluation. While recognizing the difficulty of finding (large-scale) datasets for sequential  regression tasks,  reviewers suggest running the methods on benchmark datasets. One reviewer also points out the need to compare to efficient ensemble methods for uncertainty quantification.

The AC also read  the manuscript. The AC appreciates the strong performance of the proposed method. However, given the limited number of datasets this method is evaluated, and the lacking of analysis of understanding why the method is successful or could fail, the AC recommends Reject.  The author(s) could improve the manuscript by following the suggestion of the reviewers, as well as a more detailed analysis of how robust the proposed method is: there are many parameters tuned in the empirical results and it is not clear why sensitive those parameters are.